# Accurate Determination of the Degree of Deacetylation of Chitosan Using UPLC–MS/MS

**DOI:** 10.3390/ijms23158810

**Published:** 2022-08-08

**Authors:** Ting Xue, Wenqing Wang, Zhiyuan Yang, Fanjun Wang, Lei Yang, Jian Li, Hui Gan, Ruolan Gu, Zhuona Wu, Guifang Dou, Zhiyun Meng

**Affiliations:** Beijing Institute of Radiation Medicine, Beijing 100850, China

**Keywords:** chitosan, degree of deacetylation determination, UPLC–MS/MS, relative response intensity of the characteristic peak

## Abstract

The mole fraction of deacetylated monomeric units in chitosan (CS) molecules is referred to as CS’s degree of deacetylation (DD). In this study, 35 characteristic ions of CS were detected using liquid chromatography–electrospray ionization–mass spectrometry (LC–ESI–MS/MS). The relative response intensity of 35 characteristic ion pairs using a single charge in nine CS samples with varying DDs was analyzed using 30 analytical methods. There was a good linear relationship between the relative response intensity of the characteristic ion pairs determined using ultrahigh performance (UP) LC–MS/MS and the DD of CS. The UPLC–MS/MS method for determining the DD of CS was unaffected by the sample concentration. The detection instrument has a wide range of application parameters with different voltages, high temperatures, and gas flow conditions. This study established a detection method for the DD of CS with high sensitivity, fast analysis, accuracy, stability, and durability.

## 1. Introduction

Chitosan (CS) is a linear polymer formed by the de-N-acetylation of chitin and randomly arranged and combined by β-(1–4) glycosidic bond-linked N-glucosamine (GlcN, D) units and mainly β-(1–4) glycosidic bond-linked N-acetyl-glucosamine (GlcNAc, A) units [1,2,3]. Appendix A shows the structural (reported in “Appendix A”). CS is a biodegradable and good biocompatible polysaccharide with basic character, which can be dissolved in 1% acetic or hydrochloric acid. It is popularly used in biomedicine [4,5]. The physicochemical and biological properties of CS are affected by its molar mass and degree of deacetylation (DD).

The DD of CS refers to the percentage of deacetylated glucosamine residue in CS molecules to the total glucosamine residue in the CS molecules, which is a mole fraction of deacetylated monomeric units (which can be expressed in fractions or percentages). Generally, chitin with >50% DD is known as CS, and the DD directly affects the solubility, crystallinity, viscosity, ion exchange capacity, tension, flocculation ability, immune activity, and amino-related chemical reactions of CS [6], which is an important index of CS.

Currently, there are multiple methods for determining the DD of CS, including elemental analysis [7], conductometric titration [8], potentiometric titration [9], proton nuclear magnetic resonance (^1^H NMR) [10], carbon-13 (^13^C) NMR [11], infrared spectroscopy [12,13], X-ray diffraction [14], thermal analysis [15], gas chromatography [16], ultraviolet spectroscopy [17], and nitrite deamination [18]. Each method offers both advantages and disadvantages. Generally, it is believed that ^1^H NMR is the most accurate method for determining the DD of CS [19], because it is unaffected by the accuracy with which CS samples are weighed. Moreover, the pH of the sample solution before and after detection does not change and can be repeatedly determined. Therefore, ^1^H NMR is a standard analytical technique for determining the DD of CS. The results obtained in the study of different methods for determining the DD of CS were compared with results obtained from a well-known technique, ^1^H NMR spectroscopy, to determine the accuracy of the established method.

Mass spectrometry (MS) is an effective and quantitative technique for structural analysis. It determines the mass-to-charge (*m*/*z*) ratio of compounds. Matrix-assisted laser desorption/ionization time-of-flight (MALDI-TOF) MS is a popular technology for profiling chitooligosaccharide (CS oligosaccharide, COS), which is a CS oligomer [20,21]. The conventionally used matrix is 2,5-dihydroxybenzoic acid (DHB). Both DHB and COS are soluble in water or an acetic acid solution and can be identified when mixed and detected. Currently, MALDI-TOF is popular for the structural elucidation of COS. Chen et al. [20] used MALDI-TOF MS to analyze the distribution of N-acetyl-D-glucosamine and deacetylated glucosamine units in COS with degrees of polymerization (DPs) in the range of 5–12. Bahrke et al. [22] employed MALDI-TOF technology to detect and analyze the sequence of COS. The CS sample comprises molecules with different DPs; its molecular weight typically ranges from tens of thousands to several hundred thousand. Therefore, this method is difficult for CS detection and analysis. Electrospray ionization (ESI) tandem TOF MS (or quadrupole mass analyzer, Q) is another commonly used method for analyzing COS [21,23,24]. Cord Landwehr et al. [23] used liquid chromatography (LC)–MS to determine the sequence of N-acetyl-D-glucosamine and deacetylated glucosamine units in partially acetylated COS; however, the DP range was 1–6. CS is a polymer comprising molecules with varying DPs and charged ions that are complexly formed in the mass spectrometer. Therefore, there are only a few studies on the qualitative and quantitative analysis of CS using MS.

Despite having a very high molecular weight, CS molecules may be detected and exacted by mass measurement using in-source collision-induced dissociation and form multicharged ions in the ESI source. Moreover, CS is a polymer comprising a series of molecules with different molecular weights, and these molecules comprise the GlcN and GlcNAc units. Therefore, the charged ions formed during MS may exhibit a certain regularity, and their response values may be related to the DD, molecular weight, and concentration of CS samples. Ultrahigh performance (UP)LC –MS/MS with an ESI source is a sensitive and rapid quantitative technique [25,26,27,28] and is suitable for CS characterization. This study explored the regularity of the charged CS ions formed in the ESI source. The feasibility of detecting and analyzing CS using MS is discussed, and an MS method used to analyze the DD of CS with high sensitivity and good reliability is established.

Our previous study discovered that chitosan was completely fragmented in the ESI source, and the obtained relative abundances of *m*/*z* 323.17, 484.22, 645.28, 806.33, and 967.20 in chitosan samples were ~100.00:63.12:43.85:27.04:11.82 (RSD, 7.42–15.97%). Therefore, it was preliminarily believed that chitosan has a stable MS fragmentation regularity in the ESI source, and the response value of these characteristic ion pairs has an obvious correlation with the DD detected using ^1^H NMR spectroscopy and acid-base titration [29]. Therefore, it is expected to develop a detection method. This study discovered that the addition of the responses of the characteristic ions with the same DP among the characteristic ions cleaved in the chitosan source, ∑DP1: ∑DP2: ∑DP3: ∑DP4: ∑DP5: ∑DP6: ∑DP7, have a stable ratio relationship. This confirms that chitosan is stably cleaved in the ESI source. Using the relative response intensity (RRI) of a series of characteristic ions with in-source cleaved to establish a linear relationship with the DD of CS, a UPLC-MS/MS method is developed for detecting the DD of CS. Furthermore, the detection method is confirmed from the aspects of linearity, accuracy, repeatability, stability, reproducibility, and durability. Thus, a novel detection method is established and developed for detecting DD of CS.

## 2. Results

### 2.1. Mass Spectral Characteristics of Different CSs in Q-TOF

Our previous study observed that the decomposition of CS in the ESI source produced a series of characteristic ion peaks of multimers with different DPs formed by the combination of GlcN (D) and GlcNAc (A) (Figure 1) [29]. Although the physicochemical parameters (Mw and DD) of the nine CS samples differed, they produced the same characteristic ion peaks of MS. For example, the single-charged ions of the trimer formed were 484.21 *m*/*z* ([D_3_A_0_+H]^+^), 526.22 *m*/*z* ([D_2_A_1_ + H]^+^), 568.24 *m*/*z* ([D_1_A_2_ + H]^+^), and 610.25 *m*/*z* ([D_0_A_3_ + H]^+^). Table 1 presents the combination forms of other multimers and *m*/*z* of the single-charged ions. The single-charged ion produced by the decomposition of CS in the ESI source can be calculated using Equation (1) [29].
*m*/*z* = m·D + n·A + 0/1·H_2_O + H + (m,n ∈ N).(1)

Because of the easily-broken glycosidic bonds of CS under a current strong capillary ESI-MS voltage condition, a series of characteristic ion peaks of single charges of multimers with different DPs were produced and detected using MS detectors. Figure 2j shows the possible unit combination forms of the different DPs of CS. In our previous study, the ion intensities of the nine CS samples decreased with an increase in the characteristic ion *m*/*z*, and virtually no ion signal was detected when the mass-charge ratio exceeded 1200 *m*/*z*, indicating that CS was thoroughly decomposed in the ESI source dissociation [29]. In a follow-up study, in-source dissociation stably occurred for each CS, and CS broke into an approximate proportion of polysomes with different DPs. Then, characteristic ions with high response in the samples were selected as the primary characteristic ions for analysis. The high response of single-charged characteristic ions of polysomes comprising D and A in the range of 100–1200 *m*/*z* was studied. Moreover, the response intensity of the characteristic ion peak of CS without H_2_O was analyzed.

It was observed in the response of the characteristic ions of the CS samples that they formed the same characteristic ions; however, the ion response intensity of the characteristic ion peaks differed (Figure 1).

Unlike the physical and chemical parameters of these CS samples, compared with CSs of different DDs and the observed response intensity of i+1 characteristic ions of DP = i (i ∈ N, i > 0), the ion peak response of [D_m_A_0_ + H]^+^ (m = i) was more intense than that of [D_m_A_n_ + H]^+^ (m,n ∈ N, 0 ≤ m < i, 0 < n ≤ i, m + n = i) with the CS samples of a large DD. However, for the samples with low DDs, the intensity of the ion peak of [D_m_A_n_ + H]^+^ (m,n ∈ N, 0 ≤ m < i, 0 < n ≤ i, m + n = i) appeared stronger than those of the samples having large DDs. The ion peak response intensity of [D_m_A_0_ + H]^+^ (m = i) compared to [D_m_A_n_ + H]^+^ (m,n ∈ N, 0 ≤ m < i, 0 < n ≤ i, m + n = i) was not as strong as the latter. Concerning the response intensity difference, the response intensity of these characteristic ions may have been related to the characteristic parameters, such as the DD or Mw of CS. Hence, a correlation analysis was considered using the DD and relative intensity of [D_m_A_0_ + H]^+^–[D_m_A_n_ + H]^+^ peaks with the same DP and expanded the different DPs using characteristic ion peaks.

Furthermore, MS/MS scanning of the different characteristic ions [D_m_A_n_ + H]^+^ (m,n ∈N) was conducted in CS, thus resulting in different collision energies in the range of 0–60 eV. [D_m_A_n_ + H]^+^ did not break up at low energy, and each characteristic ion uniformly produced [D_j_A_k_ + H]^+^ (j,k ∈ N, 0 ≤ j < m, 0 < k ≤ n) when the collision energy range was 15–30 eV. Large collision energies and high responses of the characteristic ions with small DPs were observed.

### 2.2. Triple Quadrupole MS to Detect the Response Intensity of CS Characteristic Ion Pairs

In the MS scanning mode of Quan-TOF MS, different CSs produced the same characteristic ions; however, the response intensity of the characteristic ion peak differed for each CS. This was probably because the characteristic ion intensity may be related to parameters, such as the DD or Mw of CS. Therefore, a Xevo TQ-S (Waters, Milford, MA, USA) was used to detect nine CS samples with 2500 ng/mL concentrations. Moreover, 35 characteristic ion pairs (*m*/*z*) of CS (Table 1) were accurately detected from characteristic ion (*m*/*z*) to characteristic ion (*m*/*z*) and assigned low collision energy in the collision chamber to reduce the fragmentation of the characteristic ions as much as possible and enable the ions to cross the quadruple mass spectrometer to reach the detector. Table 1 presents the quantitation transition characteristic ion pairs and collision energies detected in MRM-MS mode. The correlation among the response intensity of each characteristic ion pair, DD, and Mw was analyzed. Interestingly, the results demonstrated no correlation between the ion intensity and Mw; however, there was a significant positive correlation between the characteristic ion pair response intensity and the DD of CS (Figure 2a–h).

Under the ESI source MS condition, the relative abundance of the in-source cleaved CS polysomes tended to stabilize; the abundance of the polysome was expressed by the addition of the characteristic ion ([D_m_A_n_ + H]^+^, m,n ∈ N, 0 ≤ m ≤ i, 0 ≤ n ≤ i, m + n = i) pair response intensities of the polysomes (DP = i). In different CSs, the relative abundance ratio of the polysomes of DP 1–7 was similar to 100:48:45:48:23:9:4. Figure 2i shows the ion response intensity and relative abundance of each polysome in different CSs. Under ESI source parameter conditions, the breakage regularities of the different CSs were stable. This stable breakage phenomenon may be related to the glycosidic bond energy and the probability of chain-breaking to form fragment ions of different DPs under ion source bombardment conditions. The chemical bond energies of the three glycosidic bonds in CS, including GlcN–GlcN, GlcN–GlcNAc, and GlcNAc–GlcNAc were ~130 kJ/mol [30]. Therefore, the breakage of glycosidic bonds was more complete under high voltages in the ESI source, and the three glycosidic bonds were randomly occurring. Hence, along with the influence of in-source parameter settings, charged ions with different DPs could be obtained as per the stable relative abundance ratio. The stable breakage of CS under particular MS conditions provides great potential for the quantitative analysis of CS using MS.

To confirm the stability of the breakage regularity of different CSs under specific ESI source parameter conditions, repetitive experiments were designed and prepared for six 2500 ng/mL samples in parallel to examine the precision of the response intensity of 35 characteristic ion pairs of CS. The results demonstrated that the 35 characteristic ion pairs of CS exhibited good repeatability under MS conditions, and the relative standard density (RSD) was <11.08%. Figure 2k,l presents the precision results of the nine CSs in detecting the 35 characteristic ion pairs. Thus, CS exhibited stable breakage under the current ESI source condition.

### 2.3. Linear Relationship between the Relative Response Intensity (RRI) of the Characteristic Ion Pairs and the DD of CS

Under the premise of the stable breakage of CS in the current ESI source condition, it is practically significant to compare the response intensities of 35 characteristic ion pairs of different CS samples. It was observed that, unlike the mass spectra of the CS samples with different DDs in the ESI source (Figure 1), the response intensity of the [D_m_A_0_ + H]^+^ ion peak was strong for CS600-90 with a high DD, whereas that of the [D_m_A_n_ + H]^+^ peak was weak. The [D_m_A_n_ + H]^+^ peak response of CS 500-70 with a low DD appeared to be enhanced, compared with that of CS 600-90 with a high DD. Therefore, to analyze the correlation, the relative intensity between the [D_m_A_0_ + H]^+^–[D_m_A_n_ + H]^+^ peaks and the DD to analyze the correlation was used.

The response intensity of 35 characteristic ion pairs with a charge in the nine CSs with different DDs was analyzed using 30 relative ion intensity analysis methods (Table 2). Those that calculated a characteristic ion peak of polysome with DP = i (i∈N, i > 0) were defined as the ratios of [D_m_A_n_ + H]^+^ (m,n ∈ N, 0 ≤ m ≤ i, 0 ≤ n ≤ i, m + n = i) to the sum response of i + 1 characteristic ion peaks of the polysomes with DP = i using Equation (2);
(2)$$\mathrm{RRI}=[D_{m}A_{n}+H{]}^{+}\;{}_{\left(\mathrm{DP}=i\right)}\;/\;\sum_{m=i,n=0}^{m=0,n=i}[D_{m}A_{n}+H{]}^{+}\;{}_{\left(\mathrm{DP}=i\right)},$$
or those that calculated the ratios of the D response making a weighted summation according to the proportion of D in i + 1 characteristic ion peak response of DP = i (i ∈ N, i > 0) to the sum response of i + 1 characteristic ion peaks of polysomes with DP = i using Equation (3).
(3)$$\mathrm{RRI}=\sum_{}^{weight\;of\;D}\;[D_{m}A_{n}+H{]}^{+}\;{}_{\left(\mathrm{DP}=i\right)}\;/\;\sum_{m=i,n=0}^{m=0,n=i}[D_{m}A_{n}+H{]}^{+}\;{}_{\left(\mathrm{DP}=i\right)}.$$

This method was used to calculate the relative value of the characteristic ion response intensity of DP1–7 of CS and named the RRI. The RRI of each ion pair was linearly fitted with the DD of CS measured using ^1^H NMR, with the DD as the abscissa and RRI of a characteristic ion pair of CS as the ordinate in the curve. Figure 3a–g shown the results, while Table 2 presents the linear regression equation and correlation coefficient (R squared, R^2^) between the RRI of characteristic ion pairs and the DD. In particular, excellent linearity was observed between the relative response of the characteristic ion pairs measured using UPLC–MS/MS and the DD of CS. The R^2^ of the linear fitting methods was approximately 0.9–0.99. Further analysis revealed that the higher the characteristic ion pair response, the more accurate the RRI analysis, and the better the linear relationship between the characteristic ion RRI and DD. The linear relationship of the eight RRI analysis methods marked *** was the best among the 30 methods, and the R^2^ of the linear fitting exceeded 0.99. Therefore, the follow-up study will focus on these eight methods. After fitting the standard curve with this analytical method, the DD of the unknown CS samples was determined.

Eight RRI analysis methods marked *** shown in Table 2 were used to establish a standard curve for detecting different CS DDs using UPLC–MS/MS, and an equation for the linear relationship between the RRIs of characteristic ion pairs and CS DD was obtained (Table 2). Using the calculated DDs of nine CS samples, the accuracy of this method was evaluated by the relative errors (RE) between the calculated DD and that measured using ^1^H NMR. As shown in Figure 3h and Appendix A, the RE ranged from −1.93% to 2.21%. The absolute RE values were below 5%. Therefore, this method exhibited good accuracy in determining the DD of CS and can be used to determine the DD of unknown CS samples. Thus, the DD of the unknown CS samples was determined after applying the analytical method fitting to the standard curve. Under MS conditions, the eight RRI analysis methods marked with *** exhibited good repeatability, RSD was between 0.12% and 6.79%, and Figure 3i shows the RRI precision.

As per the LC–MS/MS of nine CS standards with different DDs, the RRI of characteristic ion pairs and the DD were linearly fitted to obtain a standard curve with good linearity. Using the standard curve, the calculation equation of the DD of the CS was obtained, expressed as follows (Equations (2)–(4)), where k is the slope of the linear relationship between the RRI of the characteristic ion pairs and the DD, and f is the intercept,
DD (%) = k × RRI + f.(4)

The calculation Equations (5)–(12) of the DD of CS were obtained by applying the RRI of the eight “***” methods marked as follows:DD (%) = 114.67 × D/(D + A) − 14.04,(5)
DD (%) = −114.67 × A/(D + A) + 100.63,(6)
DD (%) = 61.1 × D_2_A_0_/(D_2_A_0_ + D_1_A_1_ + D_0_A_2_) + 44.68,(7)
DD (%) = 97.81 × (D_2_A_0_ + 1/2 × D_1_A_1_)/(D_2_A_0_ + D_1_A + D_0_A_2_) + 7.04,(8)
DD (%) = 74.47 × D_4_A_0_/(D_4_A_0_ + D_3_A_1_ + D_2_A_2_ + D_1_A_3_ + D_0_A_4_) + 52.97,(9)
DD (%) = 82.2 × D_5_A_0_/(D_5_A_0_ + D_4_A_1_ + D_3_A_2_ + D_2_A_3_ + D_1_A_4_ + D_0_A_5_) + 48.33,(10)
DD (%) = 197.38 × (D_5_A_0_ + 4/5 × D_4_A_1_ + 3/5 × D_3_A_2_ + 2/5 × D_2_A_3_ + 1/5 × D_1_A_4_)/(D_5_A_0_ + D_4_A_1_ + D_3_A_2_ + D_2_A_3_ + D_1_A_4_ + D_0_A_5_) − 61.99,(11)
DD (%) = 78.86 × D_6_A_0_/(D_6_A_0_ + D_5_A_1_ + D_4_A_2_ + D_3_A_3_ + D_2_A_4_ + D_1_A_5_ + D_0_A_6_) + 48.48.(12)

Therefore, the RRI of the characteristic ion pairs of the CS sample can be measured using triple quadrupole MS, and the DD of CS can be calculated using the DD (%) equation. The simplest and most accurate method is to detect the characteristic ion pairs of [D + H]^+^ 162.08 *m*/*z* → 162.08 *m*/*z* and [A + H]^+^ 204.09 *m*/*z* → 204.09 *m*/*z* of CS and use Equation (6) or Equation (7) to calculate the DD of the CS sample to be tested. The RE of the accuracy of the measured values using this method ranged from −1.80% to 1.47%.

The mean and standard deviation of the DD of the nine CSs, measured six times with the established UPLC–MS/MS method, are shown Appendix A, the relative standard density (RSD) of the precision results was below 1.88%. The range of RE in the accuracy of the values measured using this method compared with the reference method (^1^H NMR) is shown in Figure 3h and Appendix A. The linear range of the DD of CS detected using this method is 64.5–95.2%, and the method’s detection limit is 64.5%.

### 2.4. The RRI of Characteristic Ions under Different Concentrations of the Same CS Sample Is a Fixed Value

An experiment was designed to prepare different concentrations of CS 200-87 (500, 750, 1000, 2000, and 4000 ng/mL). The samples with different concentrations were detected using the same method to analyze the RRI. The results indicated that the analyzed RRI of the characteristic ion pairs were extremely close under the same LC–ESI–MS/MS condition, and the RSD of the eight RRIs of the characteristic ion pairs marked “***” of the same CS of different concentrations was in the range of 0.40–8.95%. Then, the different concentrations of detection experiments in other CS samples were used, and the experimental results were observed. This indicated that the UPLC–MS/MS method for analyzing the DD of CS will not be affected by the accuracy of the sample concentration in the preparation process because the sample concentration will not affect the RRI of the characteristic ion pairs of CS (Figure 4).

### 2.5. Effect of the RRI of CS Characteristic Ion Pairs on the Detection Method of Different MS Parameters (Capillary Voltage, Gas Flow, and Temperature)

The primary factors affecting the formation of charged ions in ESI sources are the capillary voltage, high temperature, and nitrogen. The ionization degree of the sample can be changed by adjusting the voltage, temperature, and gas flow in the ESI source [31,32]. Therefore, the parameters, such as the capillary voltage, desolvation temperature, desolvation gas flow, cone gas flow, and nebulizer gas flow in the ESI source within a range, were adjusted. Subsequently, the RRI of 35 characteristic ion pairs of CS samples was detected.

The capillary voltage in the range of 0.2–5.0 kV was adjusted. In this process, it was observed that the capillary voltage of the ESI source was the major factor affecting the in-source ionization of CS. Using the Xevo TQ-S system (Waters, Milford, MA, USA), with an increase in the capillary voltage, the response intensity of characteristic ion pairs of CS significantly increased and then decreased when the capillary voltage was >1.0 kV (Figure 5a), but the RRIs only slightly changed (Figure 5d). Although the RRIs of the CS characteristic ions slightly changed at different capillary voltages, the relationship between the RRIs and DD at each capillary voltage was analyzed. It was discovered that they were all linear (Appendix A). Therefore, at different capillary voltages, the RRI of the characteristic ion pairs of CS is still linearly correlated with the DD.

The desolvation temperatures were adjusted in the range of 300–400 °C. With an increase in temperature, the response intensity of characteristic ions of CS increased slightly (Figure 5b), but the change in the RRI was not obvious (Figure 5e). The RRI and DD relationships at each temperature were analyzed, and it was observed that they were all linear.

Subsequently, the gas flow conditions were adjusted by setting the desolvation gas flow to 600–700 L/h, cone gas flow to 150–250 L/h, and nebulizer gas flow to 5.5–6.5 bar. By detecting the CS samples after adjusting the parameters mentioned above, the gas flow conditions had a minimal effect on the characteristic ion response intensity of CS, and the response intensity slightly increased with an increase in the desolvation gas flow; as the cone gas flow increased, the ion response intensity increased slightly, but the change was not obvious. Furthermore, with an increase in the nebulizer gas flow, the ion response intensity decreased slightly (Figure 5c). With changes in the gas flow conditions, the RRI of the characteristic ion pairs of CS remained unchanged (Figure 5f), and the relationship between the RRI and DD under each air-flow condition was linear.

This revealed that, under different MS parameter conditions, the RRI of the characteristic ions of CS only changed slightly or did not change because of the change in the ionization degree at different voltages, high temperatures, and air-flow conditions. Thus, the response of each ion changed, and the different characteristic ions had different changes in degree. Therefore, it may lead to a slight change in the RRI of each ion. Nevertheless, there was a linear relationship between the RRI of characteristic ion pairs and the DD of different CS samples under the conditions of each voltage, high temperature, and gas flow (Appendix A). Thus, the method of detecting the DD of CS using the analytical principle of the RRI of characteristic ions exhibited excellent stability and a wide range of instrument parameters (Table 3), indicating that it is a potential and promising detection method for CS DD.

The established methods were linear across a wide range of parameters, indicating that the RRI analysis method for the characteristic ions to detect the DD of CS exhibited good stability. However, because different MS parameters slightly influenced the response intensity of the characteristic ions, the method that produced the highest response was the optimal method for detecting the DD of CS. Furthermore, it supports the future quantitative detection of CS using MS. Table 3 shows the optimized parameters.

### 2.6. Method Reproduction on Different LC-MS/MS Instruments

After confirming the aforementioned experimental method, this study wanted to determine that this was not an accidental phenomenon on an MS instrument and expected that this method had good test stability and reproducibility when used on different instruments. Therefore, detection methods were developed on the SCIEX Triple Quad™ 6500plus (SCIEX AB, Framingham, MA, USA) and API4000 MS (SCIEX AB, Framingham, MA, USA) systems, with the expectation that the linear relationship between the RRI of the characteristic ion pairs and DD could be reproduced. Table 3 shows the detection method.

The methods mentioned above were confirmed on three MS instrument models under different laboratory conditions, all of which exhibited a good linear relationship, and the reproducibility of the constructed method was good. Because the detection sensitivity, resolution, and scanning range of different instruments differed, the concentration range of the analytical samples was accordingly adjusted and optimized. Table 3 shows the applicable MS parameter and detection sample concentration ranges of this method. Appendix A show the linear relationships between the RRI of the characteristic ion pairs of CS and the DD through detection on SCIEX Triple Quad™ 6500plus (SCIEX AB, Framingham, MA, USA) and API4000 MS (SCIEX AB, Framingham, MA, USA) MS systems.

## 3. Discussion

The theoretical basis for determining the DD of CS using ^1^H NMR is based on the difference between the chemical shift of the anomeric proton of the glucosamine unit (H-1D) in the deacetylated monomeric unit and the chemical shift of methyl-hydrogen in acetyl (HAc) in the acetylated monomeric unit. The proportion of the deacetylated monomeric unit in the addition of the deacetylated and acetylated monomeric units of the entire CS is calculated from the peak intensity of both [19].

The DD of CS can be analyzed using the established RRI analysis method of the characteristic ion pairs to theoretically detect the polysome with DPs of 1–7 or higher DP. This study confirmed the characteristic ions of the polysome with DPs of 1–7 and observed that the linearity was good. However, detecting ions with large *m*/*z* may easily affect the precision and accuracy, because the response intensity of the characteristic ion with large *m*/*z* is considerably lower than that of an ion with a small *m*/*z*. Larger CS-characteristic ions can be detected if the mass spectrometer used for detection has sufficient sensitivity and resolution. It was sufficient for any relative response ratio methods developed in this study to achieve an accurate and rapid analysis of the DD of CS, and the analysis method was extremely stable.

The same UPLC–MS/MS may be suitable for analyzing the DD of chitin; however, it is necessary to select a liquid phase separation method suitable for chitin. Furthermore, this idea may be useful for analyzing the parameters for characterizing the content of a characteristic monomeric unit in other macromolecular polymers comprising certain monomeric units such as the degree of vulcanization for glycosaminoglycan, mannuronate/guluronate ratio for alginates, and degree of esterification for pectin.

The advantages of this new detection method are as follows: accurate (comparable to ^1^H NMR); fast and efficient (the detection time of a single sample is 1 min), a stable method can directly refer to the results of the standard curve and test an unknown sample using the equation operation of the method; a wide measuring range and less sample consumption, the sample preparation concentration and sample injection volume are small (500–4000 ng/mL and 2 μL); economical, cheaper than high temperature ^1^H NMR; stable and durable, detection methods can be developed in different MS instruments, and suitable MS conditions have several parameters and excellent stability.

## 4. Materials and Methods

### 4.1. Materials

Nine commercial CS samples were purchased from different suppliers. CS 500-70 (448869-50 g), CS 600-66 (448877-50 g), and CS 1100-65 (419419-50 g) were purchased from Sigma-Aldrich (Darmstadt, Germany). CS 300-92 (C105801-100 g, <200 mPa·s); CS 500-78 (C105802-100 g, 200–400 mPa·s); CS 600-90 (C105803-100 g, >400 mPa·s); and CS 300-95 (C105799-100 g, 100–200 mPa·s, DD > 95%) were purchased from Aladdin (Shanghai, China). CS 200-87 (69047438-500 g, 50–800 mPa·s, DD of 80–95%) was purchased from Sinopharm Chemical Reagent Co., Ltd. (Beijing, China). CS 400-85 (C804730-100 g, high viscosity, >400 mPa·s) was purchased from Maclin (Shanghai, China). Table 4 lists the additional details of the DD, molar mass, and radius of the gyration data for these CS samples. Acetic acid, sodium chloride, and ammonium chloride were purchased from Sinopharm Chemical Reagent Co., Ltd. (Beijing, China) and were of analytical grade. High-performance liquid chromatography (HPLC)-grade acetonitrile and formic acid (85%) were obtained from Fisher Scientific (Waltham, MA, USA). Deuterium oxide and acetic acid-D4 were commercially obtained from Adamas-beta (Shanghai, China) and Sigma-Aldrich (Darmstadt, Germany), respectively. Purified water (3 ppb, <18 MΩ) was obtained from a Milli-Q water pure system (Darmstadt, Germany).

### 4.2. M_W_ Determination of CS Using Size Exclusion Chromatography Multiple Angle Laser Light Scattering (SEC-MALLS)

This study analyzed nine CS samples using an Agilent 1200 HPLC system (Agilent Technologies, Santa Clara, CA, USA) for size exclusion chromatography (SEC) coupled with a DAWN (Wyatt, Santa Barbara, CA, USA) MALLS and an Optilab (Wyatt, Santa Barbara, CA, USA) refractive index detector (RID). A Thermo MAb Pac TM SEC-1 (5 μm 300 Å, 7.8 × 300 mm, Thermo Scientific, Milford, MA, USA) gel chromatographic column was placed in an oven maintained at 25 °C, separating with a mobile phase of 100 mM NaCl–100 mM NH_4_Cl aqueous solution. The pH was adjusted to 3.3 with acetic acid. The eluent flow rate was 0.5 mL/min, and the autosampler temperature was controlled in the range of 25 ± 2 °C. The injection volume was 100 μL, and the constant mobile phase was eluted for 40 min. The molecular weight and root mean square (rms) radius moments of chitosan were determined using SEC–MALLS method. Table 4 shows Mw, Mn, Mw/Mn and R_g,z_.

### 4.3. CS DD Determination by ^1^H NMR

The CS samples were dissolved in 2% acetic acid-D4/deuterium oxide solution (*v*/*v*) at 5 mg/mL. Using a Bruker 600 MHz Avance 600 spectrometer (Bruker BioSpin GmbH, Billerica, MA, USA), ^1^H NMR spectra were recorded at 323.1 K (50 °C). The other parameters of the instrument were optimized according to [33]. The DD of CS was calculated according to Equation (13).
DD (%) = 100 × H-1D/(H-1D + HAc/3).(13)

### 4.4. LC-MS/MS

#### 4.4.1. LC

CS was separated and detected on an Acquity UPLC BEH C18 column (1.7 μm, 2.1 × 50 mm, Waters, Milford, MA, USA) and XBridge BEH C18 column (5 μm, 130 Å, 2.1 × 100 mm, Waters, Milford, MA, USA). The column temperature was thermostated at 40 °C, injection volume was 2 μL, and the mobile phase flow rate was 0.4 mL/min. The elution gradient of the mobile phase was 95% solvent A (0.1% formic acid in water)–5% solvent B (0.1% formic acid in acetonitrile), and the elution time was 1 min.

#### 4.4.2. TOF MS

The Synapt G2-Si Q-TOF MS system (Waters, Milford, MA, USA) was used to profile 10 μg/mL CS to detect direct injection. The instrument parameters were set as follows: injection rate, 15 μL/min; detection ion mode, positive; capillary voltage, 2.4 kV; cone voltage, 60 V; source offset voltage, 50 V; desolvation temperature, 350 °C; ESI source temperature, 150 °C; cone gas flow, 150 L/h; desolvation gas flow, 650 L/h; nebulizer, 6.0 bar; MS acquisition range, 50–1200 *m*/*z*; acquisition, rod-shaped; and acquisition time, 30 s. Before sample detection, the MS system was corrected using a leucine enkephalin standard and sodium formate (Waters, Milford, MA, USA). The data acquisition and processing were completed using Masslynx 4.1 software (Waters, Milford, MA, USA) [29].

#### 4.4.3. Triple Quadrupole MS

The CS samples (2500 ng/mL) were eluted and detected using an Acquity I-class UPLC tandem Xevo TQ-S MS system (with an ESI source, same as that for the Synapt G2-Si Q-TOF MS system) (Waters, Milford, MA, USA). The instrument system was controlled using Masslynx 4.2 software (Waters, Milford, MA, USA). The CS was eluted with an Acquity UPLC BEH C18 column (1.7 μm, 2.1 × 50 mm, Waters, Milford, MA, USA). The determination ion mode was positive, and the ESI source parameters were set similar to the TOF MS. The collision gas was Ar, the detection function was the multiple reaction monitoring (MRM) mode, and the collision energy was 5 eV. Table 1 shows the MRM ion pair transitions and CS collision energies.

Next, 2500 ng/mL CS samples were eluted and detected using an ExionLC™ AC UPLC-MS/MS system (SCIEX Triple Quad™ 6500plus, SCIEX AB, Framingham, MA, USA). The data were acquired and processed using Analyst 1.7.2 software (SCIEX AB, Framingham, MA, USA). The samples were eluted with an Acquity UPLC BEH C18 column (1.7 μm, 2.1 × 50 mm, Waters, Milford, MA, USA). The determination ion mode was positive, and the ESI source parameters were set as follows: collision gas (CAD), 8; curtain gas (CUR), 35 psi; ion source gas 1 (GS1), 50 psi; ion source gas 2 (GS2), 50 psi; ion spray voltage (IS), 5500.0 V; temperature (TEM), 550 °C, interface heater (ihe), on; compound declustering potential (DP), 60.0; entrance potential (EP), 10.0; collision energy (CE), 5.0; collision cell exit potential (CXP), 10.0; MRM, positive ion mode; and duration, 1.00 min. The MRM ion pair transitions of the CS were the same as those detected in the Xevo TQ-S MS (Table 1).

Then, 10 μg/mL CS samples were detected using a Nanospace HPLC system (Shiseido, Toyko, Japan) coupled with an API4000 tandem MS system (SCIEX AB, Framingham, MA, USA). The instrument system was acquired and processed using the Analyst 1.6.3 software (SCIEX AB, Framingham, MA, USA). The CS samples were eluted and separated using an XBridge BEH C18 column (5 μm, 130 Å, 2.1 × 100 mm, Waters, Milford, MA, USA). The ESI source parameters were set as follows: CAD, 6; CUR, 25 psi; GS1, 50 psi; GS2, 50 psi; IS, 5000.0 V; and TEM, 500 °C. Other parameters were set the same as those of the SCIEX Triple Quad™ 6500plus MS system.

## 5. Conclusions

CS with different DDs produced the same characteristic ion peak in MS; however, the response of the characteristic ion peak was different among varying CS samples. Nine CS samples whose concentrations were 2500 ng/mL were detected using triple quadrupole MS. There was a similar relative abundance (the response ratio of the characteristic ion pairs of the polysome that is the sum of each DP 1–7 formed by CS fragmentation under certain MS conditions), indicating that CS was stably broken into polysomes with different DPs according to a certain probability distribution under the high voltage given by particular MS conditions. The possible form of each DP combination of polysomes corresponded to the combination regularity of binary units. The characteristic ion RRI among the same DPs was closely related to the DD of CS due to the different contents of the deacetylated and acetylated monomeric units in CS macromolecular polymers.

This study detected the response intensity of 35 singly-charged characteristic ion pairs of nine CS samples with different DDs using UPLC-MS/MS and analyzed the RRIs. In the linear fitting results between the value analysis method and the DD of CS, there were eight particularly good RRIs of characteristic ion pairs, and the equations for the DD of CS after detecting the characteristic ion pair RRI of CS using LC–MS were obtained. The most accurate method was to detect the characteristic ion pair RRIs of [D+H]^+^ 162.08 *m*/*z* → 162.08 *m*/*z* and [A+H]^+^ 204.09 *m*/*z* → 204.09 *m*/*z* of CS and Equation (5) was used to calculate the DD of the CS sample to be tested.

The UPLC–MS/MS method for determining the DD of CS was unaffected by the sample concentration. The detection instrument had a wide range of applicable parameters, and there was a linear relationship between the RRI and DD at different voltages, high temperatures, and gas flow conditions.

## Figures and Tables

**Figure 1 ijms-23-08810-f001:**
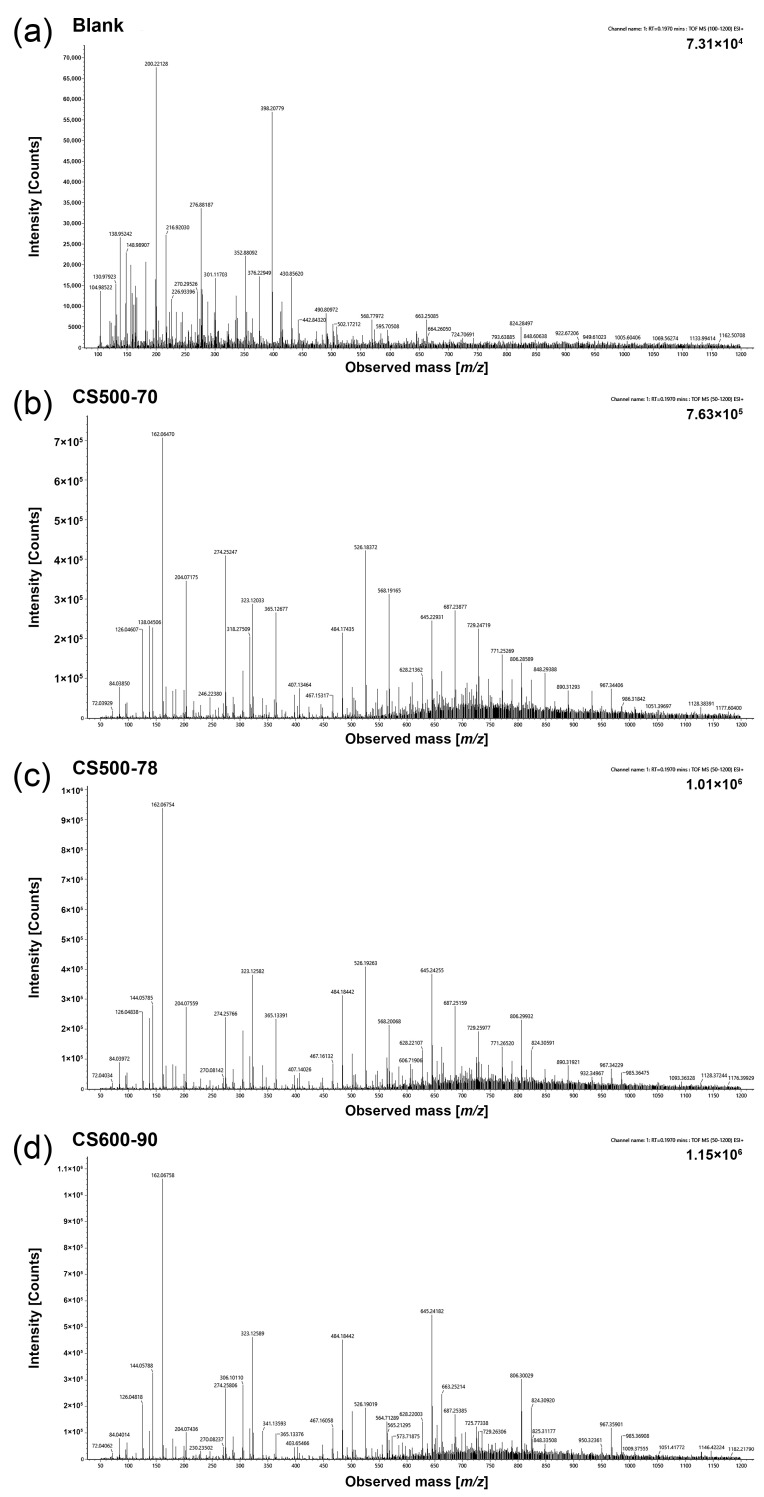
Time-of-flight (TOF) mass spectral feature map. (**a**) Blank, (**b**) CS500-70, (**c**) CS500-78, and (**d**) CS600-90.

**Figure 2 ijms-23-08810-f002:**
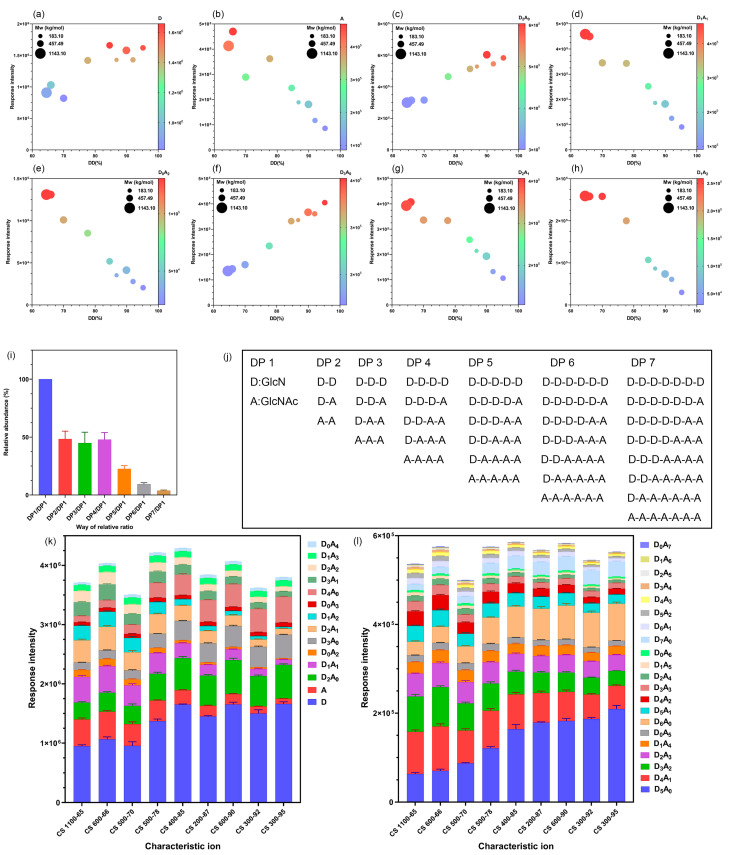
(**a**–**h**) Relationship among the ion intensity, degree of deacetylation (DD), and molecular weight (Mw) of chitosan (CS). (**i**) Ion response intensity and relative abundance of each polysome in different CSs. (**j**) Polysome combination by D and A; the relative abundance between the polysomes of the degree of polymerization (DP) 1–7 tended to be constant, i.e., DP 1:DP 2:DP 3:DP 4:DP 5:DP 6:DP 7 were relatively stable, with possible forms of D and A; D_2_A_0_, D_1_A_1_, and D_0_A_2_; D_3_A_0_, D_2_A_1_, D_1_A_2_, and D_0_A_3_; D_4_A_0_, D_3_A_1_, D_2_A_2_, D_1_A_3_, and D_0_A_4_; D_5_A_0_, D_4_A_1_, D_3_A_2_, D_2_A_3_, D_1_A_4_, and D_0_A_5_; D_6_A_0_, D_5_A_1_, D_4_A_2_, D_3_A_3_, D_2_A_4_, D_1_A_5_, and D_0_A_6_; and D_7_A_0_, D_6_A_1_, D_5_A_2_, D_4_A_3_, D_3_A_4_, D_2_A_5_, D_1_A_6_, and D_0_A_7_, respectively. The overall response of each DP is the addition of the response intensities of all possible monomeric unit forms of the DP. (**k**,**l**) Response intensity and precision results of nine CSs in detecting 35 characteristic ion pairs.

**Figure 3 ijms-23-08810-f003:**
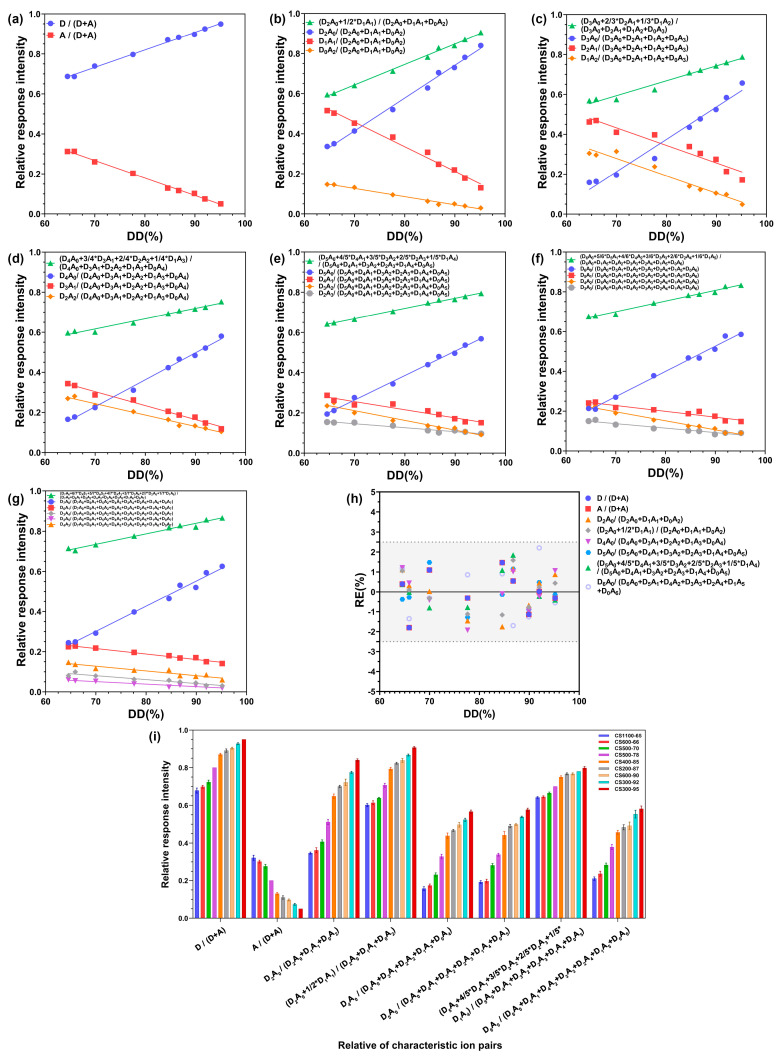
Correlation between the degree of deacetylation (DD) and relative response intensity (RRI). (**a**) DP 1, (**b**) 2, (**c**) 3, (**d**) 4, (**e**) 5, (**f**) 6, and (**g**) 7. (**h**) The relative errors (RE) between the degree of deacetylation (DD) of chitosan calculated using the established standard curve and that measured using ^1^H NMR. The area between the grey dashed lines illustrates RE ± 2.5%. (**i**) RRI precision of the characteristic ion pairs of chitosan.

**Figure 4 ijms-23-08810-f004:**
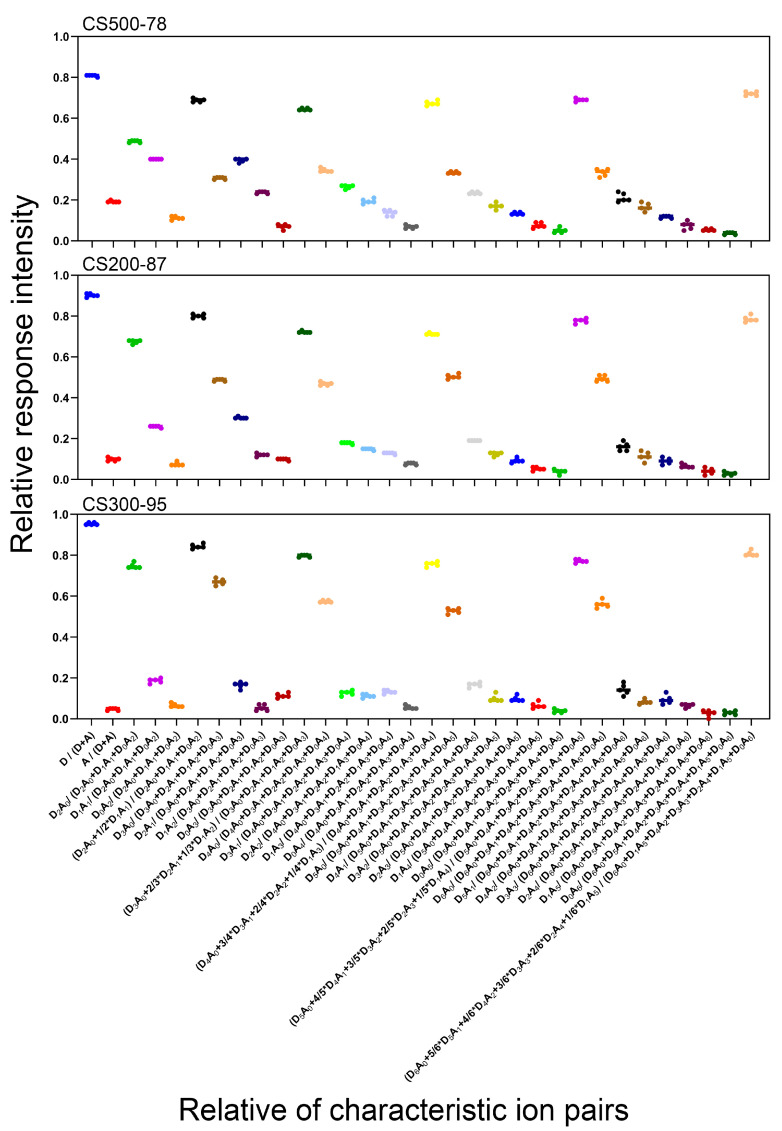
Relative response intensity (RRI) of CS 500-78, CS 200-87, and CS 300-95 with different concentrations in the range of 500–4000 ng/mL.

**Figure 5 ijms-23-08810-f005:**
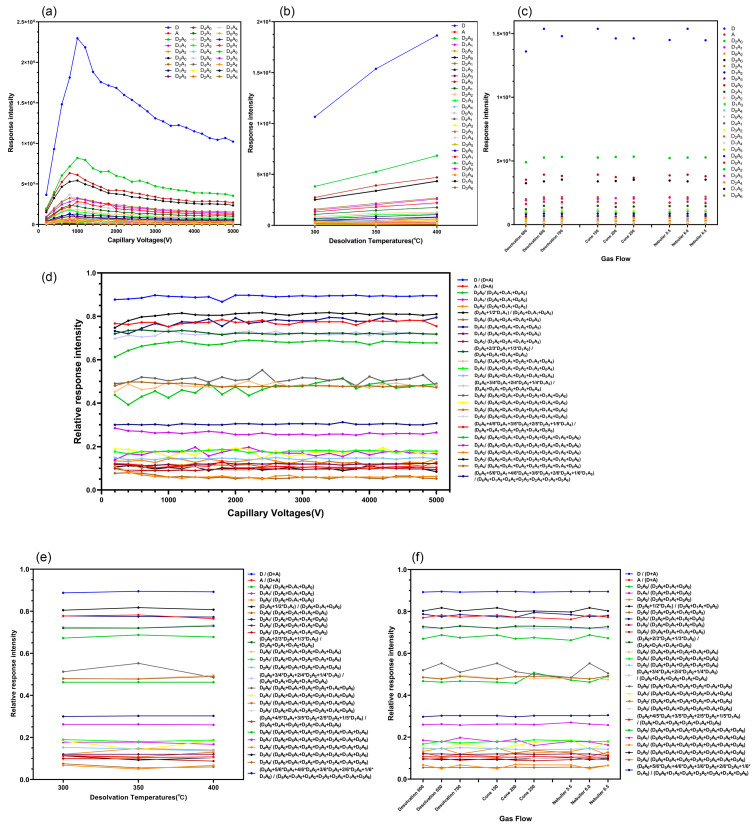
(**a**–**c**). Response intensity of 27 of the 35 characteristic ion pairs detected of CS200-87 under different mass spectrometry parameter conditions. (**d**) Relationship between Relative response intensity (RRI) and each capillary voltage from 200 to 5000 V; (**e**) desolvation temperature, 300–400 °C; (**f**) desolvation gas flow, 600–700 L/h; cone gas flow, 150–250 L/h, and nebulizer gas flow, 5.5–6.5 bar.

**Table 1 ijms-23-08810-t001:** Combination form of multimers produced by the decomposition of chitosan (CS) in the ESI source and the mass-to-charge ratio of the single-charge ions. Multiple reaction monitoring (MRM) transitions and collision energies of CS in the Xevo TQ-S mass spectrometry system.

DP	Multimer Combination	Single-Charge Ion	Characteristic Ion (*m*/*z*)	Quantitation Transition	Collision Energy (eV)
				Precursor Ion→Product Ion (*m*/*z*)	
1	D:GlcN	[D+H]^+^	162.08	162.08→162.08	5
	A:GlcNAc	[A+H]^+^	204.09	204.09→204.09	5
2	D_2_A_0_	[D_2_A_0_+H]^+^	323.15	323.15→323.15	5
	D_1_A_1_	[D_1_A_1_+H]^+^	365.16	365.16→365.16	5
	D_0_A_2_	[D_0_A_2_+H]^+^	407.17	407.17→407.17	5
3	D_3_A_0_	[D_3_A_0_+H]^+^	484.21	484.21→484.21	5
	D_2_A_1_	[D_2_A_1_+H]^+^	526.22	526.22→526.22	5
	D_1_A_2_	[D_1_A_2_+H]^+^	568.24	568.24→568.24	5
	D_0_A_3_	[D_0_A_3_+H]^+^	610.25	610.25→610.25	5
4	D_4_A_0_	[D_4_A_0_+H]^+^	645.28	645.28→645.28	5
	D_3_A_1_	[D_3_A_1_+H]^+^	687.29	687.29→687.29	5
	D_2_A_2_	[D_2_A_2_+H]^+^	729.30	729.30→729.30	5
	D_1_A_3_	[D_1_A_3_+H]^+^	771.31	771.31→771.31	5
	D_0_A_4_	[D_0_A_4_+H]^+^	813.33	813.33→813.33	5
5	D_5_A_0_	[D_5_A_0_+H]^+^	806.35	806.35→806.35	5
	D_4_A_1_	[D_4_A_1_+H]^+^	848.36	848.36→848.36	5
	D_3_A_2_	[D_3_A_2_+H]^+^	890.37	890.37→890.37	5
	D_2_A_3_	[D_2_A_3_+H]^+^	932.38	932.38→932.38	5
	D_1_A_4_	[D_1_A_4_+H]^+^	974.39	974.39→974.39	5
	D_0_A_5_	[D_0_A_5_+H]^+^	1016.40	1016.40→1016.40	5
6	D_6_A_0_	[D_6_A_0_+H]^+^	967.42	967.42→967.42	5
	D_5_A_1_	[D_5_A_1_+H]^+^	1009.43	1009.43→1009.43	5
	D_4_A_2_	[D_4_A_2_+H]^+^	1051.44	1051.44→1051.44	5
	D_3_A_3_	[D_3_A_3_+H]^+^	1093.45	1093.45→1093.45	5
	D_2_A_4_	[D_2_A_4_+H]^+^	1135.46	1135.46→1135.46	5
	D_1_A_5_	[D_1_A_5_+H]^+^	1177.47	1177.47→1177.47	5
	D_0_A_6_	[D_0_A_6_+H]^+^	1219.48	1219.48→1219.48	5
7	D_7_A_0_	[D_7_A_0_+H]^+^	1128.49	1128.49→1128.49	5
	D_6_A_1_	[D_6_A_1_+H]^+^	1170.50	1170.50→1170.50	5
	D_5_A_2_	[D_5_A_2_+H]^+^	1212.51	1212.51→1212.51	5
	D_4_A_3_	[D_4_A_3_+H]^+^	1254.53	1254.53→1254.53	5
	D_3_A_4_	[D_3_A_4_+H]^+^	1296.53	1296.53→1296.53	5
	D_2_A_5_	[D_2_A_5_+H]^+^	1338.54	1338.54→1338.54	5
	D_1_A_6_	[D_1_A_6_+H]^+^	1380.55	1380.55→1380.55	5
	D_0_A_7_	[D_0_A_7_+H]^+^	1422.56	1422.56→1422.56	5

**Table 2 ijms-23-08810-t002:** Thirty relative ion intensity analysis methods of chitosan (CS) characteristic ion pairs. The linear regression equation and correlation coefficient (R^2^) between the relative response intensity (RRI) of the CS characteristic ion pairs and the degree of deacetylation (DD) of CS. The meaning of the comment label in the table: *** R^2^ > 0.99, ** 0.9 < R^2^ < 0.99. The DD and RRI had 30 good linear relationships, and the R^2^ of linear fitting was >0.9. There were eight excellent linear relationships, and the R^2^ of the linear fitting exceeded 0.99. In the follow-up experiments, these eight analysis methods will be examined.

DP	Relative Response Intensity Ways of Characteristic Ion Pairs	Equation	R^2^	Remark
1	D/(D + A)	Y = 0.008723*X + 0.1223	0.9953	***
	A/(D + A)	Y = −0.008723*X + 0.8777	0.9953	***
2	D_2_A_0_/(D_2_A_0_ + D_1_A_1_ + D_0_A_2_)	Y = 0.01637*X − 0.7313	0.9942	***
	D_1_A_1_/(D_2_A_0_ + D_1_A_1_ + D_0_A_2_)	Y = −0.01228*X + 1.318	0.9889	**
	D_0_A_2_/(D_2_A_0_ + D_1_A_1_ + D_0_A_2_)	Y = −0.004082*X + 0.4129	0.9868	**
	(D_2_A_0_ + 1/2*D_1_A_1_)/(D_2_A_0_ + D_1_A_1_ + D_0_A_2_)	Y = 0.01022*X − 0.07196	0.9955	***
3	D_3_A_0_/(D_3_A_0_ + D_2_A_1_ + D_1_A_2_ + D_0_A_3_)	Y = 0.01623*X − 0.9241	0.9760	**
	D_2_A_1_/(D_3_A_0_ + D_2_A_1_ + D_1_A_2_ + D_0_A_3_)	Y = −0.008817*X + 1.050	0.9334	**
	D_1_A_2_/(D_3_A_0_ + D_2_A_1_ + D_1_A_2_ + D_0_A_3_)	Y = −0.008707*X + 0.8883	0.9627	**
	(D_3_A_0_ + 2/3*D_2_A_1_ + 1/3*D_1_A_2_)/(D_3_A_0_ + D_2_A_1_ + D_1_A_2_ + D_0_A_3_)	Y = 0.007454*X + 0.07184	0.9742	**
4	D_4_A_0_/(D_4_A_0_ + D_3_A_1_ + D_2_A_2_ + D_1_A_3_ + D_0_A_4_)	Y = 0.01343*X − 0.7113	0.9948	***
	D_3_A_1_/(D_4_A_0_ + D_3_A_1_ + D_2_A_2_ + D_1_A_3_ + D_0_A_4_)	Y = −0.006953*X + 0.7904	0.9872	**
	D_2_A_2_/(D_4_A_0_ + D_3_A_1_ + D_2_A_2_ + D_1_A_3_ + D_0_A_4_)	Y = −0.005676*X + 0.6412	0.9839	**
	(D_4_A_0_ + 3/4*D_3_A_1_ + 2/4*D_2_A_2_ + 1/4*D_1_A_3_)/(D_4_A_0_ + D_3_A_1_ + D_2_A_2_ + D_1_A_3_ + D_0_A_4_)	Y = 0.005125*X + 0.2580	0.9797	**
5	D_5_A_0_/(D_5_A_0_ + D_4_A_1_ + D_3_A_2_ + D_2_A_3_ + D_1_A_4_ + D_0_A_5_)	Y = 0.01217*X − 0.5880	0.9963	***
	D_4_A_1_/(D_5_A_0_ + D_4_A_1_ + D_3_A_2_ + D_2_A_3_ + D_1_A_4_ + D_0_A_5_)	Y = −0.004052*X + 0.5397	0.9390	**
	D_3_A_2_/(D_5_A_0_ + D_4_A_1_ + D_3_A_2_ + D_2_A_3_ + D_1_A_4_ + D_0_A_5_)	Y = −0.004835*X + 0.5495	0.9686	**
	D_2_A_3_/(D_5_A_0_ + D_4_A_1_ + D_3_A_2_ + D_2_A_3_ + D_1_A_4_ + D_0_A_5_)	Y = −0.001975*X + 0.2844	0.9421	**
	(D_5_A_0_ + 4/5*D_4_A_1_ + 3/5*D_3_A_2_ + 2/5*D_2_A_3_ + 1/5*D_1_A_4_)/(D_5_A_0_ + D_4_A_1_ + D_3_A_2_ + D_2_A_3_ + D_1_A_4_ + D_0_A_5_)	Y = 0.005067*X + 0.3140	0.9950	***
6	D_6_A_0_/(D_6_A_0_ + D_5_A_1_ + D_4_A_2_ + D_3_A_3_ + D_2_A_4_ + D_1_A_5_ + D_0_A_6_)	Y = 0.01268*X − 0.6147	0.9905	***
	D_5_A_1_/(D_6_A_0_ + D_5_A_1_ + D_4_A_2_ + D_3_A_3_ + D_2_A_4_ + D_1_A_5_ + D_0_A_6_)	Y = −0.002922*X + 0.4328	0.9233	**
	D_4_A_2_/(D_6_A_0_ + D_5_A_1_ + D_4_A_2_ + D_3_A_3_ + D_2_A_4_ + D_1_A_5_ + D_0_A_6_)	Y = −0.004521*X + 0.5133	0.9836	**
	D_3_A_3_/(D_6_A_0_ + D_5_A_1_ + D_4_A_2_ + D_3_A_3_ + D_2_A_4_ + D_1_A_5_ + D_0_A_6_)	Y = −0.002248*X + 0.2947	0.9374	**
	(D_6_A_0_ + 5/6*D_5_A_1_ + 4/6*D_4_A_2_ + 3/6*D_3_A_3_ + 2/6*D_2_A_4_ + 1/6*D_1_A_5_)/(D_6_A_0_ + D_1_A_5_ + D_4_A_2_ + D_3_A_3_ + D_2_A_4_ + D_1_A_5_ + D_0_A_6_)	Y = 0.005395*X + 0.3218	0.9874	**
7	D_7_A_0_/(D_7_A_0_ + D_6_A_1_ + D_5_A_2_ + D_4_A_3_ + D_3_A_4_ + D_2_A_5_ + D_1_A_6_ + D_0_A_7_)	Y = 0.01255*X − 0.5774	0.9865	**
	D_6_A_1_/(D_7_A_0_ + D_6_A_1_ + D_5_A_2_ + D_4_A_3_ + D_3_A_4_ + D_2_A_5_ + D_1_A_6_ + D_0_A_7_)	Y = −0.002736*X + 0.4073	0.9742	**
	D_4_A_3_/(D_7_A_0_ + D_6_A_1_ + D_5_A_2_ + D_4_A_3_ + D_3_A_4_ + D_2_A_5_ + D_1_A_6_ + D_0_A_7_)	Y = −0.002362*X + 0.2933	0.9032	**
	D_3_A_4_/(D_7_A_0_ + D_6_A_1_ + D_5_A_2_ + D_4_A_3_ + D_3_A_4_ + D_2_A_5_ + D_1_A_6_ + D_0_A_7_)	Y = −0.001948*X + 0.2171	0.9287	**
	D_2_A_5_/(D_7_A_0_ + D_6_A_1_ + D_5_A_2_ + D_4_A_3_ + D_3_A_4_ + D_2_A_5_ + D_1_A_6_ + D_0_A_7_)	Y = −0.001252*X + 0.1388	0.9185	**
	(D_7_A_0_ + 6/7*D_A_6_1_ + 5/7*D_A_5_2_ + 4/7*D_A_4_3_ + 3/7*D_3_A_4_ + 2/7*D_2_A_5_ + 1/7*D_1_A_6_)/(D_7_A_0_ + D_6_A_1_ + D_5_A_2_ + D_4_A_3_ + D_3_A_4_ + D_2_A_5_ + D_1_A_6_ + D_0_A_7_)	Y = 0.005193*X + 0.3718	0.9797	**

**Table 3 ijms-23-08810-t003:** The suitable range of mass spectrometry parameters and sample concentration range, and optimal parameters for the DD detection of chitosan using LC-MS/MS. The instrument parameters and sample concentration ranges of the three mass spectrometers are summarized: Xevo TQ-S; SCIEX Triple Quad™ 6500plus, and API4000.

Mass Spectrometry Model	ESI Source Parameter Category	Optimal Conditions	Parameter Range
Xevo TQ-S	Capillary Voltages (kV)	2.4	1.0–3.5
	Desolvation Temperatures (°C)	350	300–400
	Desolvation Gas Flow (L/h)	650	600–700
	Cone Gas Flow (L/h)	150	150–250
	Nebulier Gas Flow (bar)	6.0	5.5–6.5
	Concentration (ng/mL)	2500	500–4000
SCIEX Triple Quad™ 6500plus	Curtain Gas (CUR, psi)	35	30–35
	Collision Gas (CAD)	8	6–8
	IonSpray Voltage (IS, V)	5500	4500–5500
	Temperature (TEM, °C)	550	450–550
	Ion Source Gas 1 (GS1, psi)	50	40–55
	Ion Source Gas 2 (GS2, psi)	50	40–60
	Concentration (ng/mL)	2500	500–4000
API4000	Curtain Gas (CUR, psi)	25	25
	Collision Gas (CAD)	6	6
	IonSpray Voltage (IS, V)	5000	4000–5500
	Temperature (TEM, °C)	500	350–550
	Ion Source Gas 1 (GS1, psi)	50	40–55
	Ion Source Gas 2 (GS2, psi)	50	40–65
	Concentration (ng/mL)	10,000	1000–50,000

**Table 4 ijms-23-08810-t004:** Molecular characteristics and physicochemical parameters of chitosan (CS) samples.

Sample	DD (%)	Mw (kDa)	Mn (kDa)	Mw/Mn	R_g,z_ (nm)
CS 1100-65	64.5	1143.1	667.6	1.71	108.0
CS 600-66	65.9	624.6	388.7	1.61	84.6
CS 500-70	70.0	545.3	313.4	1.74	74.4
CS 500-78	77.6	496.4	342.1	1.45	67.7
CS 400-85	84.6	445.1	349.5	1.27	69.9
CS 200-87	86.8	183.1	141.9	1.29	41.6
CS 600-90	89.9	596.4	482.4	1.24	78.2
CS 300-92	92.0	290.2	215.7	1.35	56.6
CS 300-95	95.2	313.1	241.7	1.30	62.1

## Data Availability

Not applicable.

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
