# Peer review of "Accurate Determination of the Degree of Deacetylation of Chitosan Using UPLC–MS/MS"

_ijms, 2022, doi:10.3390/ijms23158810_

Round 1

Reviewer 1 Report

The literature on the analysis of the deacetylation degree of chitosan includes many papers. Determinations of this type were carried out by various methods, including mass spectrometry. Despite this, the data presented by the authors allow us to look at the problem from a different perspective and extend the spectrum of methods with a new proposal. The manuscript has been carefully prepared in terms of content and editing.

This is not, however, a critical remark. I believe that the manuscript may be accepted for publication in the proposed form.

Reviewer 2 Report

The authors present a well written study with a series of characteristic ions to monitor degree of deacetylation of chitosan.

Minor comments:

Line 28: consider adding “in supplemental information” or found in “Appendix A”, to guide the reader when finding Figure A.1a.

Line 93: consider reordering the table and table names, so that the first table referenced in the manuscript is table 1, rather than starting with introducing Table 4 as the first table here at line 93.

Lines 86, 99-108: please add the citations whenever referencing other studies such “In our previous study” and “in a follow-up study”.

Author Response

Response to Reviewer 2 Comments

Thank you for taking time out of your busy schedule to review the manuscipt. Now we have carefully corrected and replied the manuscipt for this revision. The revision instructions are as follows:

Point 1: The authors present a well written study with a series of characteristic ions to monitor degree of deacetylation of chitosan.

Response 1: Thank you very much for reviewing my manuscript in your busy schedule. Thank you for your comments and suggestions, and, sincerely, for your affirmation of the UPLC-MS/MS method for measuring the deacetylation degree of chitosan established in our research.

Point 2: Line 28: consider adding “in supplemental information” or found in “Appendix A”, to guide the reader when finding Figure A.1a.

Response 2: Thank you very much for your careful review and advice, which was very helpful for the improvement of our manuscript. As suggested by the reviewer, we added “found in Appendix A” to guide the readers when finding Figure A.1a. Corresponding corrections in revised manuscript are marked in red, shown in Line 31-32.

Point 3: Line 93: consider reordering the table and table names, so that the first table referenced in the manuscript is table 1, rather than starting with introducing Table 4 as the first table here at line 93.

Response 3: Thank you very much for your careful review and advice. This suggestion is very important. We reordered the tables and table names in the manuscript so that the first table cited in the manuscript is Table 1. Corresponding parts in revised manuscript are marked in red, shown in Line 119.

Point 4: Lines 86, 99-108: please add the citations whenever referencing other studies such “In our previous study” and “in a follow-up study”.

Response 4: Thank you very much for your careful review and advice. We apologize for the missing of relevant references. In the revised manuscript, we have added citations when citing other studies. Corresponding parts in the manuscript are marked in red, shown in Line 112-115, 126-129, and the details are as follows:

  1. Li, J.; Chen, L.; Meng, Z.; Dou, G., Development of a mass spectrometry method for the characterization of a series of chitosan. Int J Biol Macromol 2019, 121, 89-96.

Thanks again to the Editor and Reviewers for your hard work! Best wishes to you!

Reviewer 3 Report

Major points:

11)      Since the authors have already published a paper on the use of UPLC-ESI-MS/MS for the characterization of chitosan, the novelty of this study and how this work differs from the previous one should be clearly and explicitly described in the Introduction.

22)      Since the authors propose a new method for determining DD of chitosan, it is advisable to compare the analytical characteristics (mean, standard deviation, linear range, detection and quantification limits) of the designed method (UPLC-ESI-MS/MS) with the reference method (1H NMR). I really missed this comparison in the study.

33)      Table 3: What is Rg,z(nm)? Is it the radius of gyration of chitosan aggregates? How was it measured and what is its role in this study?

44)      The English writing style could have been better. There are too many English errors and wrong constructions, which sometimes drastic affect the scientific meaning, making the paper difficult of reading and understanding. A native English speaker with a scientific background should carefully revise the manuscript prior its resubmission.

Minor points:

15)      Line 11-12 and 33-35: DD is not a percentage, but a mole fraction of deacetylated monomeric units (which can be expressed in fractions or percentages). Please correct.

26)      Line 20: I would remove "inexpensiveness" because the UPLC-ESI-MS/MS method is quite expensive and comparable in cost to 1H NMR.

37)      Line 29: Replace “alkaline polysaccharide polymer” with “cationic polysaccharide” or “polysaccharide with basic character”. Alkaline is an adjective for alkali and its use here is incorrect.

48)      Line 35: Replace “55% DD” to “50% DD”

59)      Line 85-86: Remove this sentence, which is a repetition of the other one (lines 25-27).

610)      Lines 86, 99 and 191: Add appropriate references after “In our previous study”.

711)      The molecular weight (in contrast to the molar mass) is unitless. Either remove kg/mol (Table 3) or use term molar mass across the text.

812)      Line 359: The “amino-hydrogen (H1D)” is not visible in 1H NMR spectrum of chitosan in D2O. Apparently, you were referring to the anomeric proton (H-1) of the glucosamine unit. Please, correct.

913)      You often use the term monomer to refer to a monomeric unit, which is not quite accurate. Please correct throughout the text.

Author Response

Response to Reviewer 3 Comments

Thank you for taking time out of your busy schedule to review the manuscipt. Now we have carefully corrected and replied the manuscipt for this revision. The revision instructions are as follows:

Point 1: 11) Since the authors have already published a paper on the use of UPLC-ESI-MS/MS for the characterization of chitosan, the novelty of this study and how this work differs from the previous one should be clearly and explicitly described in the Introduction.

Response 1: Thank you for your careful review and professional advice. Actually, in our published paper, it was found that chitosan was completely fragmented in the ESI source, we preliminarily believe that chitosan has a stable mass spectrometry fragmentation regularity in the ESI source, and the response value of these characteristic ion pairs has obvious correlation with the degree of deacetylation. Therefore, it is expected to develop a detection method. In comparison, in this study, we further prove that chitosan is stably cleaved in the ESI source. Using the relative response intensity (RRI) of a series of characteristic ions with in-source cleaved to establish a linear relationship with the degree of deacetylation of chitosan, we developed a UPLC-MS/MS method for the detection of the degree of deacetylation of chitosan. And the detection method is verified from the aspects of linearity, accuracy, repeatability, stability, reproducibility and durability. Thus, a novel detection method is established, and a new detection way is developed for the detection of chitosan deacetylation degree.

We agree with the reviewer that the differences between this study and the previous one should be clearly described in the introduction section to highlight the novelty of this study. Therefore, according to your suggestion, we have made changes at the end of the introduction and referenced and updated our published literature on the use of UPLC-ESI-MS/MS for the characterization of chitosan to compare and highlight our research aim and innovations with the available current literature. Corresponding parts in the manuscript are marked in red, shown in line 93-108.

Point 2: 22) Since the authors propose a new method for determining DD of chitosan, it is advisable to compare the analytical characteristics (mean, standard deviation, linear range, detection and quantification limits) of the designed method (UPLC-ESI-MS/MS) with the reference method (1H NMR). I really missed this comparison in the study.

Response 2: Thank you for your professional advice. We have carefully considered the comparison characteristics you provided, which are very important to us. Actually, we did a large amount of comparative data in the course of our research, but due to the word limitations of the manuscript, we did not expand the description in the original manuscript, and we can provide it to you if needed. According to your suggestion, we have added a paragraph in the section 2.3 and data results in the Appendix A of Table A.2.. Comparative analysis the designed method (UPLC-ESI-MS/MS) with the reference method (1H NMR), we have refined the description of the analytical properties of the former method. Corresponding parts in the manuscript are marked in red,shown in line 290-295.

Point 3: 33) Table 3: What is Rg,z(nm)? Is it the radius of gyration of chitosan aggregates? How was it measured and what is its role in this study?

Response 3: Thank you very much for your careful review. Rg, z(nm) is the root mean square radius moments, which is using SEC-MALLS method in section 4.2 to detect. The determination of such characteristic parameters as rms radius is mainly to characterize the physicochemical parameters of chitosan, so that we can clearly understand the important parameters of chitosan used in the experiment. Besides, it helps us make more accurate and precise description of the characteristics of chitosan. To make it more clearly, we have modified our manuscript in the line 484-486 of section 4.2 to supplement the description.

Point 4: 44) The English writing style could have been better. There are too many English errors and wrong constructions, which sometimes drastic affect the scientific meaning, making the paper difficult of reading and understanding. A native English speaker with a scientific background should carefully revise the manuscript prior its resubmission.

Response 4: Thank you for your careful review and pointing out our mistakes. According to your suggestion, we have carefully examined and revised the English writing of the manuscript. In addition, we submitted our revised manuscript to the editing service of the https://www.nesediting.com/ for further language editing, and the Editing Certificate we were also attached with our revised manuscript, shown in Figure 1. Corresponding corrections in the manuscript are marked in red.

Figure 1. The Editing Certificate of the revised manuscript.

Point 5: 15) Line 11-12 and 33-35: DD is not a percentage, but a mole fraction of deacetylated monomeric units (which can be expressed in fractions or percentages). Please correct.

Response 5: Thank you very much for your advice. We apologize for this misnomer. To make it more accurate, the description of DD in the manuscript was revised to mole fraction of deacetylated monomeric units. Corresponding parts in the manuscript are marked in red, shown in line 11-13 and 38-39.

Point 6: 26) Line 20: I would remove "inexpensiveness" because the UPLC-ESI-MS/MS method is quite expensive and comparable in cost to 1H NMR.

Response 6: Thank you very much for your careful review and advice. This suggestion is very important. We appreciate your professional advice. We have removed the " inexpensiveness " in line 22. Corresponding parts in the manuscript are marked in red.

Point 7: 37) Line 29: Replace “alkaline polysaccharide polymer” with “cationic polysaccharide” or “polysaccharide with basic character”. Alkaline is an adjective for alkali and its use here is incorrect.

Response 7: Thank you very much for your careful review and advice, which was very helpful for the improvement of our manuscript. As suggested by the reviewer, we replaced “alkaline polysaccharide polymer” with “polysaccharide with basic character”. Corresponding parts in the manuscript are marked in red, shown in line 33.

Point 8: 48) Line 35: Replace “55% DD” to “50% DD”.

Response 8: Thank you very much for your careful review. We have made changes to line 40. Corresponding parts in the manuscript are marked in red.

Point 9: 59) Line 85-86: Remove this sentence, which is a repetition of the other one (lines 25-27).

Response 9: Thank you very much for your careful review. We removed the sentence in line 111-112. Corresponding parts in the manuscript are marked in red.

Point 10: 610) Lines 86, 99 and 191: Add appropriate references after “In our previous study”.

Response 10: Thank you very much for your careful review. We have added references after “In our previous study” in line 112-115 and 126-129, and we have modified lines 229 of the manuscript. Corresponding parts in the manuscript are marked in red.

  1. Li, J.; Chen, L.; Meng, Z.; Dou, G., Development of a mass spectrometry method for the characterization of a series of chitosan. Int J Biol Macromol 2019, 121, 89-96.

Point 11: 711) The molecular weight (in contrast to the molar mass) is unitless. Either remove kg/mol (Table 3) or use term molar mass across the text.

Response 11: Thank you very much for your careful review. This suggestion is very important. We removed kg/mol in Table 3. Corresponding parts in the manuscript are marked in red.

Point 12: 812) Line 359: The “amino-hydrogen (H1D)” is not visible in 1H NMR spectrum of chitosan in D2O. Apparently, you were referring to the anomeric proton (H-1) of the glucosamine unit. Please, correct.

Response 12: Thank you very much for your careful review. We apologize for this mistake. In the revised manuscript, we have made corresponding modification in lines 420-421 of the manuscript. Corresponding parts in the manuscript are marked in red.

Point 13: 913) You often use the term monomer to refer to a monomeric unit, which is not quite accurate. Please correct throughout the text.

Response 13: Thank you very much for your careful review. This suggestion is very important. We have revised the full text of our manuscript thoroughly and replaced “monomer” to “monomeric unit”. Corresponding parts in the manuscript are marked in red.

Thanks again to the Editor and Reviewers for your hard work! Best wishes to you!

Round 2

Reviewer 3 Report

In the revised manuscript, the authors have successfully addressed most of the reviewers’ concerns and have made the necessary revisions. The paper has been generally improved and, in my opinion, can be published in the present form.